# SARS-CoV-2 Specific Nanobodies Neutralize Different Variants of Concern and Reduce Virus Load in the Brain of h-ACE2 Transgenic Mice

**DOI:** 10.3390/v16020185

**Published:** 2024-01-25

**Authors:** María Florencia Pavan, Marina Bok, Rafael Betanzos San Juan, Juan Pablo Malito, Gisela Ariana Marcoppido, Diego Rafael Franco, Daniela Ayelen Militelo, Juan Manuel Schammas, Sara Elizabeth Bari, William Stone, Krisangel López, Danielle LaBrie Porier, John Anthony Muller, Albert Jonathan Auguste, Lijuan Yuan, Andrés Wigdorovitz, Viviana Gladys Parreño, Lorena Itat Ibañez

**Affiliations:** 1Instituto de Química Física de los Materiales, Medio Ambiente y Energía (INQUIMAE), Universidad de Buenos Aires, Consejo Nacional de Investigaciones Científicas y Técnicas, Buenos Aires ZC 1428, Argentina; mflorenciapavan@gmail.com (M.F.P.); danielamilitello94@gmail.com (D.A.M.); sara.elizabeth.bari@gmail.com (S.E.B.); 2Incuinta, Instituto Nacional de Tecnología Agropecuaria (INTA), Hurlingham, Buenos Aires ZC 1686, Argentina; bok.marina@inta.gob.ar (M.B.); malito.juanpablo@inta.gob.ar (J.P.M.); awigdo@gmail.com (A.W.); 3Instituto de Virología e Innovaciones Tecnológicas, Consejo Nacional de Investigaciones Científicas y Técnicas (IVIT-CONICET), Hurlingham, Buenos Aires ZC 1686, Argentina; schammas.juan@inta.gob.ar; 4Instituto de Química Biológica de la Facultad de Ciencias Exactas y Naturales (IQUIBICEN), Departamento de Química Biológicas, Consejo Nacional de Investigaciones Científicas y Técnicas, Buenos Aires ZC 1428, Argentina; rafael.betanzossj@gmail.com; 5Centro de Investigaciones en Ciencias Veterinarias y Agronómicas (CICVyA), Instituto Nacional de Tecnología Agropecuaria (INTA), Hurlingham, Buenos Aires ZC 1686, Argentina; marcoppido.gisela@inta.gob.ar (G.A.M.); franco.diego@inta.gob.ar (D.R.F.); 6Department of Entomology, College of Agriculture and Life Sciences, Fralin Life Science Institute, Virginia Polytechnic Institute and State University, Blacksburg, VA 24061, USA; wstone97@vt.edu (W.S.); klopez5@vt.edu (K.L.); danip@vt.edu (D.L.P.); jmuller51@vt.edu (J.A.M.); jauguste@vt.edu (A.J.A.); 7Center for Emerging, Zoonotic, and Arthropod-Borne Pathogens, Virginia Polytechnic Institute and State University, Blacksburg, VA 24061, USA; lyuan@vt.edu; 8Department of Biomedical Sciences and Pathobiology, Virginia-Maryland College of Veterinary Medicine, Virginia Polytechnic Institute and State University, Blacksburg, VA 24061, USA

**Keywords:** SARS-CoV-2, nanobodies, virus neutralization, variants of concern, intranasal treatment

## Abstract

Since the beginning of the COVID-19 pandemic, there has been a significant need to develop antivirals and vaccines to combat the disease. In this work, we developed llama-derived nanobodies (Nbs) directed against the receptor binding domain (RBD) and other domains of the Spike (S) protein of SARS-CoV-2. Most of the Nbs with neutralizing properties were directed to RBD and were able to block S-2P/ACE2 interaction. Three neutralizing Nbs recognized the N-terminal domain (NTD) of the S-2P protein. Intranasal administration of Nbs induced protection ranging from 40% to 80% after challenge with the WA1/2020 strain in k18-hACE2 transgenic mice. Interestingly, protection was associated with a significant reduction in virus replication in nasal turbinates and a reduction in virus load in the brain. Employing pseudovirus neutralization assays, we identified Nbs with neutralizing capacity against the Alpha, Beta, Delta, and Omicron variants, including a Nb capable of neutralizing all variants tested. Furthermore, cocktails of different Nbs performed better than individual Nbs at neutralizing two Omicron variants (B.1.529 and BA.2). Altogether, the data suggest the potential of SARS-CoV-2 specific Nbs for intranasal treatment of COVID-19 encephalitis.

## 1. Introduction

In late 2019, Severe Acute Respiratory Syndrome Coronavirus 2 (SARS-CoV-2) emerged in Wuhan/China and was found to be responsible for the infectious respiratory disease COVID-19, which ultimately resulted in an unprecedented pandemic affecting millions of people and placing a significant burden on global healthcare systems and economic activities. The virus spread rapidly, reaching a fatality rate between 0.4% and 1.5% worldwide [1,2]. To date, there have been more than 772,138,818 confirmed cases and over 6,985,964 deaths reported to the World Health Organization (WHO) [3]. In Argentina, COVID-19 reached an alarming caseload with a total of 10.4 million cases and 130,463 deaths [4].

SARS-CoV-2 is a member of the *Coronaviridae* family, genus *Betacoronavirus* in the order *Nidovirales* [5]. The SARS-CoV-2 virion encapsidates a positive-stranded RNA genome protected by a nucleocapsid (N) and covered by a lipid bilayer envelope [6]. The envelope contains three different glycoproteins: the Spike protein (S), the Envelope protein (E), and the Membrane protein (M) [7]. The S proteins are organized as trimers on the viral envelope, with each monomer being composed of 1273 amino acids (AA) divided into two subunits: the S1 subunit, containing the receptor binding domain (RBD), and the S2 subunit that allows fusion of viral and cellular membranes. During infection, the RBD interacts with the Angiotensin Converting Enzyme 2 (ACE2) receptor located on the membrane of the target cells, with the S protein promoting viral fusion with host cell membranes to allow virus entry into the host cell [8].

Great efforts have been made to develop and test different types of vaccines to combat the infection. To date, there are a total of 12 vaccines commercially available [9] and 70.6% of the world population has received at least one dose of a COVID-19 vaccine [10]. However, limited immunity responses (i.e., low magnitude and short duration) to natural infection and vaccination, as well as the emergence of new SARS-CoV-2 variants of concern (VOCs), opens questions regarding the efficacy of the vaccines and highlights the need for the development of novel and more innovative prophylactic and therapeutic products [11,12].

The use of antivirals represents an alternative for SARS-CoV-2-infected people and a complementary strategy to control COVID-19. Several approaches using monoclonal (mAbs) and polyclonal antibodies or proteins, which demonstrate binding specificity and neutralizing capacities, are being explored [13,14]. Monoclonal antibodies that target the S or RBD proteins are useful in treating SARS-CoV-2 infection [13,15]. However, it has been reported that the efficacy of mAbs against specific variants and subvariants can be variable [16] and recent variants, such as the Omicron, are often unaffected by their use [17]. Even more, there is always the concern that these protein drugs may induce anti-drug antibodies (ADAs) [16,18]. The Food and Drug Administration (FDA) has granted emergency use authorization to antibody therapies, generally in the form of a mixed cocktail of antibodies, to better target a broad spectrum of SARS-CoV-2 variants [19]. For this reason, other passive immunization strategies, such as the use of llama-derived Nbs, are an attractive alternative to develop highly sensitive diagnostic methods, and contribute to the preventive and therapeutic treatment of the disease [20].

Llama-derived Nbs are composed of only the variable region (VHHs) of the heavy chain antibodies present in the serum of camelids [21]. These antibody fragments have a small molecular weight (15 kDa) and are one of the smallest molecules known in nature to have an antigen-binding function [22]. Nanobodies have high expression yields, are easy to produce, and are associated with significantly lower production costs compared to conventional mAbs [22]. More importantly, Nbs possess several unique properties compared to mAbs and derived antibody fragments, such as better stability at different pH and high temperatures, higher water solubility, and the ability to cross the blood–brain barrier (BBB) and penetrate deep into the tumors [22]. Nanobodies also have access to hidden cryptic epitopes on the surface of antigens, causing viral disassembly [23,24], and are easily modifiable to improve their binding and neutralizing properties [25]. Even more, most Nbs induce low or no ADAs, so they have broad perspectives in the fields of pharmaceutical applications, clinical diagnostics, and therapeutics [26].

Here, we report the identification and characterization of several S protein-specific Nbs that efficiently neutralize different variants of SARS-CoV-2 in vitro. Intranasal administration of Nbs induced protection against WA1/2020 challenge in k18-hACE2 transgenic mice, linked to reduced viral replication in nasal turbinates and brain. We identified a Nb, Nb-45, capable of neutralizing all variants of SARS-CoV-2 tested in its monomeric form. We also showed that combinations of two Nbs targeting different domains of the S protein enhanced their neutralizing properties against some Omicron subvariants. Altogether, the data suggest the potential of SARS-CoV-2 specific Nbs for the treatment of COVID-19.

## 2. Materials and Methods

### 2.1. Cells and Virus

HEK-293T cells were grown in high glucose Dulbecco’s modified Eagle’s medium (DMEM, Thermo Fisher Scientific, Waltham, MA, USA) supplemented with 10% fetal bovine serum (Natocor, Córdoba, Argentina), penicillin/streptomycin (Thermo Fisher Scientific, Waltham, MA, USA) and 110 mg/L of sodium pyruvate (Thermo Fisher Scientific, Waltham, MA, USA). Coronavirus Isolate hCoV-19/USA-WA1/2020 (BEI, NR-52281) and Argentinian isolate hCoV-19/Argentina/PAIS-C0102/2020 (D614G), adapted to grow in Vero cell line, were used for Virus Neutralization Assay (VNA). Isolate hCoV-19/USA-WA1/2020 was also used for mice challenge.

### 2.2. Recombinant Protein Expression and Purification

Mammalian expression plasmids encoding for SARS-CoV-2 S-2P protein containing a C-terminal 6xHis tag were kindly provided by Dr. Karin Bok (VRC7473-2019 nCoV S-dFurin-WT-F3CH2S_JSM) and Dr. Florian Krammer (pCAGGs nCoV19). The sequences were modified to remove the polybasic cleavage site and a pair of proline substitutions were performed to stabilize the prefusion conformation [27]. The RBD-expressing vector (pCAGGs nCoV19, residues 319 to 541) was kindly provided by Dr. Florian Krammer [27]. ACE2s-HRP was cloned in a modified pCAGGs vector containing a human serum albumin secretion signal and a 6xhis tag. The ACE2 sequence was cloned from the ACE2_TM_3CHS vector kindly provided by the VRC and the HRP coding sequence was added at the C-terminal. Plasmids were transiently transfected in HEK-293T cells using polyethyleneimine (PEI, PolyAR, Buenos Aires, Argentina) in a 1:2.5 DNA:PEI ratio in OptiMem medium (Thermo Fisher Scientific, Waltham, MA, USA). The supernatant containing recombinant proteins was harvested 72 h post-transfection and purified by gravity flow using Poly-Prep Chromatography Columns (Bio Rad, Hercules, CA, USA) pre-charged with nickel affinity resin (Amintra, Abcam Inc., Waltham, MA, USA). Fractions containing the recombinant proteins, as judged by the SDS-PAGE analysis, were pooled, dialyzed against phosphate buffer saline (PBS), and stored at −80 °C. SARS-CoV-2 S-2P protein expressed in a CHO-pool system was produced by Dr. Yves Durocher at the National Research Council of Canada as it has been previously described [28]. The expression of RBD in yeast (yRBD) was done by the Argentinian AntiCovid Consortium (members listed in the acknowledgments section) using the X-33 *P. pastoris* strain as described [29].

### 2.3. Llama Immunization and Library Construction

A llama located at INTA’s Camelids Experimental Unit was intramuscularly immunized according to the schedule described in Figure 1A. Three different sources of S-2P protein were used due to the low performance obtained in adherent cells (HEK-293T) and the urgency in protein availability for the immunizations. Complete Freund’s adjuvant was used for priming and Incomplete Freund’s adjuvant for the following boosters. The antibody responses to S-2P and RBD were monitored by enzyme-linked immunoassay ELISA and pseudovirus neutralization test (pVNT) on days 4 and 7 post each inoculation. Llama management, inoculation, and sample collection were conducted by trained personnel under the supervision of a Doctor in Veterinary Medicine following Argentinean and international guidelines for animal welfare. This study was approved by the Internal Committee for the Care and Use of Experimental Animals (Spanish acronym CICUAE) under protocol N° 15/2020. A Nb-library was obtained as already stated [30]. Another Nb-library, previously obtained after immunizing a llama with 4 × 10^9^ FFU/dose (fluorescent focus forming units) of BCoV Mebus strain, was also used in this study.

### 2.4. Isolation of SARS-CoV-2 S-2P- and RBD-Specific Nanobodies

Three consecutive rounds of panning were performed as previously described with minor modifications [31]. Wells of a MaxiSorp microtiter plate (Nunc, Thermo Fisher Scientific, Waltham, MA, USA) coated overnight at 4 °C with 0.2 μg of recombinant protein S-2P or RBD, were used to select specific binders. As negative control, wells were coated with supernatants of non-transfected HEK-293T cells or non-transformed yeast, respectively. Screening of specific binders was performed by ELISA on 96 randomly selected individual colonies from the second and third rounds of panning, using both periplasmic extract (PE) and recombinant phages (rP). The periplasmic extract was prepared as described [32]. To prepare samples for rP ELISA (rPE), individual colonies were grown overnight in 0.5 mL of 2×YT medium containing 100 μg/mL ampicillin and 1% glucose. After that, 5 μL of culture from each colony was diluted in 0.5 mL of fresh medium and grown for 2 h before being infected with 5 × 10^9^ VCSM13 phages/well. After 30 min of incubation without shaking, the plates were centrifuged for 15 min at 1800× *g* to remove glucose. Bacteria were grown overnight in 2 mL of 2×YT medium containing ampicillin and kanamycin. The next day, supernatants containing phages were pelleted by centrifugation for 15 min at 1800× *g*, and recombinant phages were recovered from the supernatant by PEG/NaCl precipitation.

### 2.5. Periplasmic Extract ELISA (PEE) and Phage ELISA (rPE)

Nanobodies produced from PE or as rP from individual colonies were tested for binding to either SARS-CoV-2 S-2P or RBD protein. MaxiSorp microtiter plate wells (Nunc) were coated overnight at 4 °C with 100 ng/well of recombinant proteins (S-2P and yRBD) or irrelevant proteins as negative controls. After washing with PBST, wells were blocked with 10% skimmed milk powder in PBST and 100 μL of the PE or rP diluted ¼ was added to the wells. Nanobody-specific binding was detected by PEE as previously described [33]. In the case of rPE, HRP-labeled anti-M13 phage antibody diluted 1:3000 (GE Healthcare, Piscataway, NJ, USA) was used as a detection system. To determine specific binding, the OD_405_ value of antigen-coated wells at least two times higher than the OD_405_ value of the control wells, was considered positive. Plasmids from positive clones were transformed in DH5α cells and further characterized by a restriction reaction before sending samples for sequencing.

### 2.6. Phylogenetic Analysis and Germline Origin

MEGA 11 was used to analyze genetic relatedness and clustering of the unique Nbs using a neighbor-joining tree with 1000 bootstrap replicates. The IMGT/V-QUEST program was used to determine the CDRs of the selected Nbs and to analyze the germline origin using their nucleotide sequences [34]. The IMGT/V-QUEST program, which includes only *Vicugna pacos* (alpaca) nucleotide sequences, allowed a rough analysis of the V, D, and J domains’ origins. The sequences were also analyzed using IMGT/DomainGapAlign, in which the protein sequences are compared with a *Lama glama* database. In this case, only information on the V and J domains was retrieved [35]. The sequence logo was plotted using WebLogo3 [36].

### 2.7. Expression and Purification of Nanobodies

Production and purification of the selected clones (*n* = 43) was carried out at 37 °C in TB medium supplemented with 100 μg/mL ampicillin and 0.1% glucose. Nanobody expression was induced with 1 mM IPTG for 16 h at 28 °C. Periplasmic extracts were obtained as described [30]. Nanobodies from PE were purified using IMAC Hi-Trap columns (GE Healthcare) and eluted using 300 nM Imidazole. Further purification was performed by size exclusion gel filtration using a Superdex 75 column (GE Healthcare) using AKTA Prime Plus (GE Healthcare). Nanobodies were concentrated using Vivaspin centrifugal concentrators with a cut-off of 3 kDa (Sartorius, Goettingen, Germany) and stored at −20 °C.

### 2.8. Binding Affinity to the Target Antigen Estimated by ELISA EC_50_

To evaluate the binding capacity of the selected Nbs to the S-2P and RBD proteins, MaxiSorp 96-well plates (Nunc) were coated overnight at 4 °C with 100 ng/well of recombinant proteins in carbonate/bicarbonate buffer pH 9.6. Plates were blocked with 10% skimmed milk in PBST for 1 h at 37 °C. Purified Nbs were adjusted to a concentration of 1 μM and 10-fold dilutions were added to the coated plates and incubated for 1 h at 37 °C. A homemade polyclonal rabbit serum against Nbs diluted 1:2000 was added, followed by a commercial HRP-conjugated goat anti-rabbit IgG diluted 1:5000 [37]. All incubations were performed at 37 °C. The reaction was developed with ABTS and stopped with 5% SDS, absorbance at 405 nm was measured. The EC_50_ was estimated using a four-parameter log-logistic regression model (AAT Bioquest, Inc., Pleasanton, CA, USA., Quest Graph IC_50_ Calculator) [38].

### 2.9. Pseudovirus Neutralization Test (pVNT)

Pseudovirus-expressing Wuhan-Hu-1 SARS-CoV-2 S proteins were produced by co-transfection of plasmids encoding a GFP protein (Addgene, Watertown, MA, USA, 11619), a lentivirus backbone (VRC5602, NIH, Boston, MA, USA), and the S protein (VRC7475_2019-nCoV-S-WT, NIH) in HEK-293T cells as previously described [39]. To produce pseudoviruses expressing the S protein from different variants, the following plasmids were used: Alpha (B.1.1.7), Beta (B.1.351), and Delta (B.1.617.2) (InvivoGen, San Diego, CA, USA). Plasmids encoding the S protein from the Omicron variants were obtained from the G2P-UK National Virology consortium. Triplicate two-fold serial dilutions of Nbs starting at a dilution of 1 μM, or six-fold diluted heat-inactivated llama serum (56 °C for 45 min), were prepared in 50 μL of OptiMem medium and combined with an equal volume of titrated pseudoviruses, incubated for 2 h at 37 °C, and then added to HEK-293T cells previously transfected with plasmids coding for the ACE2 receptor and TMPRSS2 protease (VRC9260, NIH). Forty-eight hours later, cells were observed under the microscope (IX-71 OLYMPUS, Breinigsville, PA, USA), and GFP-positive cells were automatically counted with ImageJ. Inhibition percentage was calculated by the following formula: 100 × [1 − (X − MIN)/(MAX − MIN)] where X stands for the number of GFP-positive cells at a given concentration of Nb, and MIN and MAX refer to the number of GFP-positive cells in uninfected cells or in cells transduced with only pseudovirus, respectively. IC_50_ titers were determined based on sigmoidal nonlinear regression using GraphPad Prism software Inc. (La Jolla, CA, USA) [40]. A two-parameter Logistic model was used with the following equation: Y = 100/(1 + 10^((LogIC50 − X) × HillSlope))). X: log of concentration; Y: normalized response; IC50: half-maximal inhibitory concentration; and Hillslope: slope factor.

### 2.10. Wild-Type Virus Neutralization Assays (VNA)

Neutralization assays were conducted using the isolate hCoV-19/Argentina/PAIS-C0102/2020 (D614G). Vero cells, 1.5 × 10^4^ cells/well were seeded in 96-well plates and incubated for 24 h at 37 °C in a 5% CO_2_ atmosphere. Two-fold diluted Nbs (starting concentration 10 μM) were incubated with 100 TCID_50_ of the virus at 37 °C for 1 h in DMEM supplemented with 2% fetal bovine serum, penicillin/streptomycin and 10 μg/mL amphotericin B. Cells were infected with virus/Nb mixture and incubated for 1 h at 37 °C and 5% CO_2_. Inoculum was washed and cells were incubated with DMEM with 2% fetal bovine serum for 72 h at 37 °C until a cytopathic effect (CPE) was observed. Cells were then fixed with 70% acetone. Virus replication was confirmed by immunofluorescent staining using a homemade fluorescein isothiocyanate (FITC)-labeled polyclonal IgG llama serum produced against SARS-CoV-2 S-2P protein. The neutralizing titer was calculated as the inverse of the highest dilution that evidences positive fluorescence, comparable to non-infected Vero cells.

A plaque reduction neutralization test (PRNT) was performed using the United States isolated (USA-WA1/2020, NR-52281). Two-fold serially diluted Nbs in DMEM medium supplemented with 2% FBS (starting concentration 1 μM) were incubated with a viral suspension containing 100 plaque-forming units of SARS-CoV-2 virus at 37 °C for 72 h. Cells were then fixed with 4% paraformaldehyde for 20 min at 4 °C and stained with crystal violet solution in methanol. The CPE was assessed visually considering minor damage to the monolayer (1–2 plaques) as well as a manifestation of CPE. Neutralization titer was defined as the highest serum dilution without any CPE in two of three replicable wells. The number of infected cells was determined per well by counting localized areas of clearance in the cell monolayer left undeveloped by the crystal violet. PRNT_50_ was calculated using the NIAID Calculator [41]. The maximal inhibitory concentration was established at 90% (IC_90_) reduction in the CPE detected at the light microscope, corresponding also to the same reduction in fluorescent focus forming units.

### 2.11. Competition of Nanobodies with ACE2

To determine whether the Nbs prevent the SARS-CoV-2-RBD/ACE2 interaction, we set up a surrogate virus neutralization test based on the ELISA technique. For this, RBD was adsorbed overnight at 4 °C in 96-well plates at a concentration of 0.2 μg/well in carbonate/bicarbonate buffer pH 9.6. Plates were washed 3 times with PBST and blocked for 1 h at room temperature (RT) with 200 μL 3% skimmed milk. Afterward, two-fold serial dilutions of Nbs (starting concentration 1 μM) were added to the wells and incubated for 1 h at RT. Recombinant ACE2-HRP was incubated for 2 h at RT. After washing, 50 μL of 3,3′, 5,5′ tetramethylbenzidine (TMB, BD) was added for 15 min. The reaction was stopped with 50 μL H_2_SO_4_ and the plates were read in a spectrophotometer at 450 nm (TECAN). The binding isotherms were analyzed by a non-linear regression model using GraphPad Prism software Inc. (La Jolla, CA, USA). A two-parameter Logistic model (2PL) was used with the following equation: Y = 100/(1 + (IC_50_/X)^HillSlope). X: concentration; Y: normalized response; IC_50_: half-maximal inhibitory concentration; and Hillslope: slope factor.

### 2.12. Interference of Nanobody Binding to Spike by Biliverdin

Nanobodies selected with the S-2P protein and which do not bind to RBD, were tested in an ELISA-based biliverdin competition assay. For this, biliverdin IX α (biliverdin) was obtained by oxidative ring opening of hemin IX (Sigma Aldrich, St. Louis, MO, USA) and purified following reported protocols [42]. Stock solution of biliverdin IX α was prepared in DMSO and diluted to 1% for working solutions. Concentration was determined by absorbance at 388 nm (ε = 39,900 M^−1^ cm^−1^). Ninety-six-well plates were coated overnight with 0.05 µg of purified S-2P protein and next day blocked with 1% skimmed milk in 0.5% PBST. Subsequently, 25 µL of 10 µM biliverdin was added to half of the wells and incubated for 5 min. Then 25 µL of ten-fold diluted Nbs were added (starting concentration 1 µM) and incubated for 1 h at RT. Specific binding was determined using an HRP-conjugated anti-HA antibody diluted 1:5000 (Abcam Inc., Waltham, MA, USA), and the reaction was developed with TMB substrate and read at 450 nm. In a second experiment, a fixed concentration of Nbs (0.1 µM) was mixed with a five-fold serial dilution of biliverdin (starting concentration 25 µM) and the reaction was developed as previously mentioned.

### 2.13. Efficacy of Nanobodies in a Mouse Model

To assess the protective efficacy of the Nbs against SARS-CoV-2 infection, 4-week-old k18-hACE2 mice (Jackson Labs, Bar Harbor, ME, USA) were separated into 7 groups (*n* = 9) of mice with approximately equal numbers of males and females in each group. Four hours before challenge, mice were administered intranasally with 10 or 20 µg of anti-rotavirus control Nb (2KD1), anti-SARS-CoV-2 Nb-39, Nb-43, Nb-45, Nb-104, or Nb-110. Mice experiments were performed at the beginning of this project when little characterization of Nbs had already been done; the selection of the Nbs to test and their concentration was mostly based on the availability and preliminary experiments. Nanobody 39 and Nb-43 were potent neutralizers as measured by pVNT and for that reason, as well as for the limited amount of purified Nb available, 10 µg of these Nbs were administered. Nanobody 145 was not tested in the mouse model due to the low availability of animals and the high similarity in its CDR3 sequence compared with Nb-39. Mice were then challenged intranasally with 1 × 10^5^ PFU of the WA1/2020 strain of SARS-CoV-2 in each nostril. This dose produces 100% lethality in K18-hACE2 as confirmed by previous studies. Mice were then monitored daily for weight loss and survival, with checks increasing to at least 3 times daily when disease symptoms presented. Four days post-challenge, 3 mice in each group (1 male, and 2 females, excluded from weight and survival data) were euthanized, and tissues (i.e., brain, lungs, and nasal turbinates) were collected to assess the impact of the Nb treatment on viral titers by RT-qPCR as previously described [43]. Primer sequences were taken from the 2019-Novel Coronavirus (2019-nCoV) Real-time rRT-PCR Panel from the Centers for Disease Control and Prevention (CDC). Quantitative synthetic SARS-CoV-2 RNA from ORF1ab, E, and N was used for the generation of a standard curve to determine viral load (ATCC, VR-3276SD). The survival data were analyzed by the Mantel–Cox log-rank test. Virus titers were log10 transformed and analyzed under a general linear mixed model analysis of variance, where treatment and tissue were considered fixed factors with interaction. The heterogeneity of variance among groups was modeled using a varIdent variance–covariance matrix. Post-ANOVA multiple comparisons of the mean virus load in each tissue inside each treatment group were analyzed by LSD Fisher test and *p*-values corrected by the Bonferroni method. The analyses were conducted in Infostat with a link to R [44]. All animal experiments and operations were performed in the biosafety level 3 (BSL-3) facility, and the protocols were approved by the Institutional Animal Care and Use Committee at Virginia Tech, institutional animal care and use committees (IACUC) number 21-065 SARS-CoV-2, Pathogenesis, and Countermeasure Testing.

### 2.14. In Silico Analysis

Nanobodies were modeled using deep learning-based end-to-end modeling, Nanonet [45]. The top-rated structures were selected and subsequently refined for 100 ns using molecular dynamics simulation with AMBER22. To maintain homogeneity, identical parameters were used for all cases (see Appendix B). VMD software was used for visualizations and image rendering.

Protein–protein docking was performed using the High Ambiguity Driven protein–protein Docking (HADDOCK) with a semi-flexible and unrestricted docking protocol [46,47]. The prefused Spike RBD up (6VYB) was used as the docking template, and multiple docking runs were performed with Nb-43, Nb-45, and Nb-53 against the S protein to identify the best result based on the lowest RMSD clustering score. Additionally, docking of the ligand biliverdin to a specific region of the S protein (AA 121–207) was carried out to evaluate its biological relevance.

To analyze the molecular interactions among the docking results, a custom Python script was developed. The script calculates the distance between the AA of the Nb and the S protein by computing the center of mass of each residue within a 7 Å distance. This generates a list of potential interacting residues. To analyze these interactions, the list of residues was imported into VMD, and the distance tool was used to determine the distance between specific AAs. The types of AA interactions were also identified and analyzed.

## 3. Results

### 3.1. Selection of SARS-CoV-2 S-2P and RBD-Specific Nanobodies

A one-year-old male llama (*Lama glama*), seronegative for antibodies to human SARS-CoV-2, was immunized with recombinant SARS-CoV-2 S-2P and RBD proteins, according to the schedule described (Figure 1A). After three immunizations, the llama developed strong antibody responses to the S-2P protein, and RBD as measured by ELISA, reaching an IgG antibody titer of 262,144 for both antigens on post-immunization day (PID) 32 (Figure 1B). The llama’s antibody response showed strong virus-neutralizing activity, measured by pVNT (Figure 1B,C). The neutralization capacity increased after each immunization, reaching a peak at a dilution of 1:1296 from a serum sample taken on PID 32 (IC_90_). After confirming optimal antibody responses, the llama rested unvaccinated for one month to promote the hypermutation process and to improve the affinity of the humoral response [48]. An immune library of 1.8 × 10^9^ independent transformants was constructed using peripheral blood lymphocytes (PBLs). All random colonies that were controlled (48/48) had a fragment of ~700 bp, indicating the incorporation of the coding sequence of a Nb (Figure 1D).

Another VHH library, previously obtained after immunizing a llama with the BCoV Mebus vaccine, developed a strong Ab response to the SARS-CoV-2 S-2P and RBD proteins (Figure 2A,B). However, the polyclonal Abs did not neutralize SARS-CoV-2 by pVNT, even when the S proteins from both strains show considerable sequence similarity (44.4%) and resemblance in overall structure (Figure 2B–D).

Three consecutive rounds of panning were performed to select specific Nbs for the S-2P and RBD proteins. After biopanning with the S-2P protein, 74 clones were positive for rPE and 45 of them recognized RBD. By PEE, 62 S-2P-specific clones were detected and 27 also bound to RBD. When the biopanning was performed with the RBD protein, rPE and PEE resulted in 46 and 53 positive clones, respectively. After sequencing, 43 unique Nbs were detected from the SARS-CoV-2-library and 2 from the Mebus-library (Mebus Nb-10M and Mebus Nb-25M) (Appendix A).

### 3.2. SARS-CoV-2 Nanobody Characterization

We constructed a phylogenetic tree with the nucleotide sequences of the selected Nbs from the SARS-CoV-2 immune library using MEGA version 11 (Appendix A). Even though some Nbs can be clustered in four groups, most of them cannot be grouped due to a large sequence variability. Results from the IMGT/V-QUEST analysis showed a predominant use of the V3S53*01 gene and allele for groups A and B, while most of the V genes for groups C and D were V3-3*01 (Appendix A). More variability in V gene usage was retrieved by the IMGT/DomainGapAlign analysis. When analyzing J segments, J4*01 was used in all groups according to the V-QUEST program. Nevertheless, for the DomainGapAlign program, J6*01 was predominant for clusters A and B, and J4*01 for C and D. D segments have great variability according to the V-QUEST program, as can be seen in Appendix A. Only Nb-30 has a different allele (in this case, D3*02). The CDR3 of the selected Nbs averaged 15 (12–20) AA long and showed high sequence diversity. Considering the Nb sequences and their germinal and phylogenetic origin, we can affirm that when screening a large Nb-library (1.8 × 10^9^ independent transformants) and using two selection antigens, it is possible to obtain Nbs with high sequence variability and diverse origin.

Twenty-nine Nbs, whose level of expression ranged from 1 to 9.2 mg/L, were selected for further characterization, and tested by ELISA against recombinant SARS-CoV-2 S and RBD to determine affinities and domain specificities (Figure 3 and Appendix A). Most of them showed half-maximal effective concentration (EC_50_) values in the single-digit-nM range, indicating strong binding to these proteins. Seven Nbs selected with the S-2P protein did not bind to RBD and only one Nb recovered from the BCoV Mebus library recognized this antigen alone and in the context of S-2P (Appendix A).

### 3.3. Screening of Neutralizing Nanobodies against SARS-CoV-2 WT Strain

A preliminary screening identified 15 potential neutralizers, inhibiting SARS-CoV-2 pseudovirus infection in a dose-dependent manner with half-maximal inhibitory concentrations (IC_50_) values ranging from 3.36 to 79.04 nM. Among these, three Nbs selected with the RBD protein and five Nbs selected against the S-2P protein showed strong neutralization activities and were chosen for further studies (Figure 4). Nanobody 10M, selected from the BCoV Mebus library, although able to recognize the RBD domain by ELISA, did not show any neutralizing property. The neutralization ability of the selected Nbs was also determined by VNA, applying two different methodologies, IF and PRNT, and two WT isolates: hCoV-19/Argentina/PAIS-C0102/2020 (containing the mutation D614G) circulating in Argentina, and hCoV-19/USA-WA1/2020 from USA. The results, summarized in Figure 4C, show the neutralization capacities exhibited by the selected Nbs. It is important to mention that even though a difference in the IC_90_ values can be observed for each assay, the same Nbs were identified to be neutralizing in two different tests, performed by two independent laboratories, and using two distinct isolates.

We next examined whether the neutralizing Nbs were able to block RBD interaction with the ACE2 receptor in a competition assay, which was measured by a surrogate ELISA (Figure 5A). Nanobody 39, Nb-43, Nb-104, Nb-110, and Nb-145 were found to compete with ACE2 for binding to RBD with an IC_50_ of 9.16, 80.21, 5.47, 15.94, and 14.83 nM, respectively. These were classified as RBD binders. Nanobody 45, Nb-51, and Nb-53 did not compete with ACE2, suggesting that they bind to epitopes outside the RBD, and were classified as non-RBD binders.

To further map the epitope recognized by the non-RBD binders, we carried out a biliverdin competition assay, as it has been described that this metabolite binds to an epitope on NTD of the S protein and competes with a fraction of S-specific serum antibodies [49]. Our results show that the addition of 5 μM biliverdin reduced the binding of Nb-45, Nb-51, and Nb-53 to the S-2P protein by a percentage of −25.19, −20.96, and −29.24, respectively. By contrast, Nb-145 binding (RBD binder) was not affected by the addition of biliverdin (4.45%) (Figure 5B). In a separate experiment, a dose–response assay was performed using Nbs at 100 nM and biliverdin from 12.5 to 0.1 μM. In this study, we confirmed that the binding of Nb-45, Nb-51, and to a lesser extent Nb-53, was reduced in the presence of biliverdin and that this decline increased with a higher concentration of this metabolite (Appendix A–C). Altogether, these results suggest that the neutralizing non-RBD binders might be detecting epitopes located close to the NTD of the S-2P protein.

### 3.4. Protection against SARS-CoV-2 Challenge in k18-hACE2 Mouse Model

To assess the protective efficacy of Nbs against SARS-CoV-2 infection and mortality, k18-hACE2 mice were challenged intranasally with 1 × 10^5^ PFU of the WT strain. Eighty percent of mice treated with Nb-39 were protected from mortality and a significant reduction in virus loads (3–4 Log10 titer) in nasal turbinates and brain sections was observed compared to the negative control group (rotavirus Nb 2KD1) (Figure 6B and Figure 7, general mixed linear model, *p* < 0.001). These results were in concordance with the high VN titer of this Nb observed in the neutralizing tests conducted in vitro. Animals receiving Nb-110, Nb-104, and Nb-43 showed survival proportions of 60%, 50%, and 40%, respectively. Moreover, Nb-104 and Nb-110 significantly reduced virus loads in nasal turbinates and the brain compared to the untreated group (Figure 7A,C). Nanobody 43 marginally reduced the virus load in all tissues. Mice treated with Nb-45 (non-RBD binder) or irrelevant rotavirus-specific Nb-2KD1 died between days 5 and 10 post-challenge (Figure 6B). We observed that all mice that were partially protected after Nb treatment either retained or increased their body weight throughout the experiment, while the control group and mice treated with Nb-45 showed a decrease in body weight (Figure 6A). Even though all mice treated with Nb-45 died, samples taken four days after challenge showed a significant reduction in virus load in the brain (Figure 7C, One-way ANOVA among treatments, in each tissue, LSD Fisher, *p* < 0.001). None of the selected Nbs showed a significant reduction in viral load in the lower respiratory tract (Figure 7B).

### 3.5. Nanobodies to SARS-CoV-2 WT Strain Neutralize SARS-CoV-2 Variants

We next studied the neutralizing breadth of RBD- and non-RBD binders against SARS-CoV-2 variants. Pseudoviruses expressing the S protein of the Alpha (B.1.1.7), Beta (B.1.351), Delta (B.1.617.2), and Omicron (B.1.1.529, BA.2, XBB, and XBB.1.5) variants were prepared. Except for Nb-43, RBD binders maintained their ability to neutralize the Alpha variant (Table 1 and Appendix A). Nanobody 104 and Nb-145 demonstrated comparable potencies and Nb-110 showed ~20-fold reductions in its neutralizing potency. Similar results were observed for the Delta variant, although neutralizing potencies diminished for all RBD binders (~25-fold). Severe reduction in or complete loss of neutralizing activities was found for these Nbs when pVNT was performed against the Beta variant. The neutralizing capacity of Nb-43 was observed only for the Omicron variants B.1.1.529 and BA.2. Analyzing the non-RBD binders, the neutralizing potencies of Nb-53 decreased between 4 and 10 times against the Alpha, Beta, and Omicron XBB subvariants, while the same Nb completely lost its neutralizing activity against Omicron BA.1 and BA.2 subvariants (Table 1 and Appendix A). Nanobody 45 and Nb-53 exhibited the broadest neutralizing activity against all variants tested. Remarkably, Nb-45 showed lower IC_50_ values compared to Nb-53 and both Nbs still maintained nearly the same neutralizing potencies against the Omicron XBB subvariants compared to the WT strain.

### 3.6. A cocktail of Nanobodies Enhances Neutralization Potencies against Omicron Variants

Finally, we decided to focus on the Nbs that were able to neutralize the Omicron variants, as these variants were recently circulating in our population. We tested a cocktail of Nb-43 (RBD binder), Nb-45, and Nb-53 (non-RBD binders). Combinations of Nb-43 and Nb-45 or Nb-45 and Nb-53 did not show any enhancement in their neutralizing capacity against the Omicron B.1.1.529 variant, but a cocktail of Nb-43 and Nb-53 significantly increased their potency (One-way ANOVA, Tukey HSD, *p* < 0.001) (Figure 8A,C,E). When Nb-45, Nb-51, and Nb-53 were mixed, a significant increase in their neutralizing capacity was observed only when compared with Nb-43 alone or the combination of Nb-43 and Nb-45 (*p* < 0.001) (Figure 8A,C,E). For the BA.2 Omicron variant, combinations of Nb-43 and Nb-45 or Nb-43 and Nb-53 significantly reduced their IC_50_ value (*p* < 0.001), suggesting a synergistic effect between two Nbs that presumably bind to different epitopes (Figure 8B,D,E). The neutralization potency increased significantly when Nb-45, Nb-51, and Nb-53 were added together compared with each Nb alone (*p* < 0.001). In contrast to the results obtained for the B.1.1.529 Omicron variant, the three-Nb mixture, compared with the combination of two Nbs, showed a significant difference in its IC_50_ value for the BA.2 Omicron variant, especially for the Nb-43 and Nb-53 or Nb-45 and Nb-53 mixtures, *p* < 0.001. Similar experiments were conducted with the XBB and XBB.1.5 Omicron variants and non-RBD binders as only those Nbs were able to neutralize these strains. No significant increase in the neutralizing capacity was observed in this case (Appendix A).

### 3.7. Nanobody Epitope Mapping and Binding Mode by In Silico Prediction

Considering that Nb-45 and Nb-53 exhibited the broadest neutralizing activity against all variants tested, these two Nbs and one RBD binder, Nb-43, were selected and modeled for further structure prediction analysis. HADDOCK provided multiple results with varying scores; we selected the best pose which positioned the Nbs’ CDRs in direct contact with the S protein (Figure 9 and Appendix A). Our analysis of the interacting residues between Nb-43 and the S protein suggested that the binding site would be located at the binding interface of the complex. Residues 469, 511, and 512 of the RBD may contribute as critical residues for binding affinity (Figure 9A). Other residues, 463 to 511, within a distance less than 5 Å, might also be interacting with the Nb residues. Preliminary studies of the Nb-43-Spike complex caused the disassembly of the protein impairing the determination of the exact epitope (Dr. Martin Hällberg, personal communication).

For the non-RBD binders (Nb-45 and Nb-53), the analysis of the predicted interactions allowed us to identify potential interacting residues in the NTD of the S protein. Residues 54 (CDR2), 101, and 114 (CDR3) of Nb-45 may be interacting with residues in the NTD at positions 60, 244, and 249, respectively (Figure 9B). In the case of the Nb-53, possible residues involved in the Nb–NTD interaction might be 54 (CDR2), 106, and 107 (CDR3); and 116, 206, and 231, respectively (Figure 9C). We also performed a HADDOCK analysis of biliverdin and observed that the predicted binding site of Nb-45 and this molecule overlap. When a similar analysis was performed for Nb-53, we observed a partial overlap with the biliverdin recognition site. Our findings support the competition assay results, but further mutagenesis analysis and crystallographic studies are needed to confirm the residues involved in the S–Nb interaction.

## 4. Discussion

Given that the COVID-19 pandemic and the emergence of new variants continue to pose a global health threat, there is still an urgent need for the development of broad-spectrum molecules to combat the disease. So far, several Nbs have been developed with promising diagnostic and therapeutic applications [20,50,51,52,53,54]. Both monomeric Nbs and engineered molecules showed strong neutralizing activity to the different VOCs that have emerged, including the Omicron subvariants [52,55,56,57,58,59,60,61]. The intense production of new recombinant Nbs highlights the relevance of this technological platform, which is expected to reach clinical use for SARS-CoV-2 and other viral diseases in the near future [62]. Here, we report the development and characterization of a novel and diverse set of unique Nbs with strong neutralizing activities. We identified seven lead molecules that were classified as RBD or non-RBD binders. Nanobody 45, a non-RBD binder, showed the broadest neutralizing activity against all SARS-CoV-2 variants tested. Nanobodies were even able to significantly reduce viral loads in the brain of challenged mice and, most importantly, neutralize some Omicron variants alone (Nb-45) or when combined in a cocktail (Nb-43, Nb-45, and Nb-53).

After an immunization protocol of four injections, an exceptionally large immune library of 1.8 × 10^9^ individual transformants was generated. We obtained different groups of Nbs that exhibited high sequence variability and originated from a diverse set of V, D, and J genes (Appendix A). Our results are consistent with those obtained by other groups screening camelid-derived libraries, where an impressive diversity in the Nb nucleotide sequence, specificity, and neutralizing capacity was found, especially after successive immunizations to promote the development of superimmunity [48,54]. This is particularly relevant considering the emergence of new VOCs, since the screening of a large library, like the one we described in this work, constructed from hyperimmunized animals, can broaden the type of Nbs obtained and increase the probability of finding binders that recognize new variants.

Network analysis showed that the Nbs are arranged into several clusters sharing different properties, such as target recognition, neutralization potency, CDR3 length, and isoelectric point (pI) (Figure 10 and Appendix A). It is worth highlighting that we isolated Nbs with similar sequence identities using two different antigens for the biopanning. Most selected Nbs (32 out of 43) recognize the RBD protein. As it was reported by others, the lower number of non-RBD binders retrieved from a VHH library is likely attributed to the highly antigenic nature of the RBD and the masking effect on non-RBD regions caused by the glycan shield of the SARS-CoV-2 spike [63,64]. The RBD binders, despite being the strongest neutralizers against the WT strain (average IC_50_: 10.44 nM), had the poorest breadth against the variants tested. Severe reduction in or complete loss of neutralizing activities was found for these Nbs when the pVNT was performed against the Beta and Omicron variants, probably due to K417N, E484K, and N501Y mutations in the RBD of the Beta variant and 22 residue substitutions of the XBB subvariants [65]. Three of the largest clusters obtained by network analysis are composed of Nbs with strong neutralizing activity (IC_50_ below 25 nM) and a wide breadth of neutralization of SARS-CoV-2 variants. This was the case with Nb-45, a non-RDB binder that might be interacting with the NTD region of the S protein as observed in competition assays (Figure 5). Neutralizing antibodies directed to this domain have also been reported by others [49,66]. Even though crystallographic and structural experiments would be needed to fully understand the Nb–NTD interaction, the epitope binding prediction suggested that Nb-45 and Nb-53 are indeed interacting with this domain (Figure 9).

Nanobodies have shown promise as prophylactic or therapeutic tools given their high specificity and low immunogenicity [26,62]. Due to their small size, Nbs have few immunogenic epitopes; they usually do not form immunogenic aggregates, and are rapidly eliminated from the blood [62]. These antibody fragments exhibit high sequence identity with the human IGHV3 family, which also contributes to their low immunogenicity. Even more, they can be further humanized [67]. As an example, a humanized anti-TNFα trivalent, bispecific Nb did not induce ADAs during long-term administration, even in an animal model of secondary failure [68]. Two non-humanized Nbs showed a low immunogenicity risk profile in humans after assessment of several parameters, such as ADA determination, aggregation analysis, and in vitro immunogenicity assays [69]. Taken together, the low immunogenic profile of Nbs makes them suitable for human and mice administration and their use against SARS-CoV-2 infection has been considered [70].

Protection after treatment with Nb-39, Nb-104, and Nb-110 was associated with a reduction in virus load in the upper respiratory tract, as well as in the brain (Figure 7 and Figure 8). In this regard, it has been reported that Nbs can cross the BBB after some modifications [71]. Furthermore, a study has revealed that Nbs with a high isoelectric point (pI~9.0) spontaneously cross the BBB [72]. Such Nbs not only gained access to the brain but were even found to penetrate cells and bind to intracellular proteins. Nanobody 45 and Nb-104 have the highest pI (8.98 and 7.98, respectively), and their potential capability to cross the BBB could explain the reduction in virus load in the brain, even when Nb-45 did not prevent mice death and did not show a significant decrease in viral titer in the lung. These results suggest that although the neutralization exerted by Nbs in other tissues could lead to an overall lower viral load, neutralization directly in the brain cannot be ruled out. The capability of Nbs to cross the BBB could be of high impact when encephalitis has been diagnosed in patients with acute and long-term COVID [73,74]. In contrast to this hypothesis, Nb-39 showed an acidic pI (pI: 5.28) and was also able to reduce the virus load in the brain. In this case, the high reduction in virus replication in the respiratory tract might reduce virus dissemination to other organs, including the brain. Although our hypothesis regarding Nbs’ access to the brain needs to be confirmed by radiolabeling biodistribution assays, our data suggest that intranasal administration of a cocktail of Nbs could be used to prevent or treat COVID-19 encephalitis. This statement is further supported by other studies that have reported that Nbs can reach the brain after intranasal administration [75,76,77].

Our Nbs were not able to reduce virus replication in the lower respiratory tract, contrasting with several examples of Nbs inhibiting SARS-CoV-2 in this tissue. Using Fc-tagged Nbs at a dosage of 20 mg/kg per mouse, a survival rate of 60 to 100% was achieved and viral loads in the lungs and nasal turbinates were significantly lower than that of the control group [78]. Another Fc-tagged Nb, administered at 10 mg/kg, significantly reduced viral load in the lower respiratory tract but failed to provide complete protection [79]. It is important to mention that most of the Nbs applied in vitro and in vivo in a mouse model were previously multimerized [55,80,81,82,83,84,85,86]. A gain in the KD value, increased affinity, and more potent neutralizing capability have been reported for Nb trimerization compared with dimerization [86,87]. Also, multimerization can enhance the neutralizing activity of individual Nbs in vitro by increasing their apparent affinity and combinations of Nbs binding to independent epitopes are more potent in preventing replication [88]. This group also demonstrated that a delay or prevention of escape mutations to biparatopic Nbs in vitro can translate into better performance in vivo [88]. In our work, only monovalent Nbs at a low dose were intranasally applied. We then expect that the application of multimerized Nbs, mono- or biparatopic, will enhance neutralization and, hopefully, more effectively inhibit SARS-CoV-2 replication in the lower respiratory tract. However, it should be considered that multimerization will increase the size of these molecules and BBB crossing could be affected.

The emergence and spread of SARS-CoV-2 variants that can escape the neutralization by mAbs pose a challenge to the development of effective vaccines and therapeutics [15]. For this reason, Nb cocktails, composed of two or more Nbs that recognize different epitopes on the S protein, have been suggested as a strategy to increase the neutralization potency and prevent viral escape [64,88]. Our study shows that the combination of an RBD-binder and non-RBD binder (Nb-45, NTD-specific Nb) enhanced the neutralizing potency against some Omicron variants (B.1.1.529 and BA.2). Zhao and collaborators have reported the development of a Nb capable of binding to both RBD and NTD of the S trimer through the same CDR3 loop, with broadly neutralizing activity [89]. These findings highlight the potential of Nbs cocktails as a strategy for developing novel prophylactic and therapeutic tools against SARS-CoV-2, particularly in the context of emerging viral variants. However, it is worth mentioning that Nb-45 alone has a broad neutralizing capacity and can inhibit the transduction of all variants tested in this work. Therefore, Nbs that bind to different epitopes, have strong neutralizing properties, and reduce the viral load in the brain represent promising tools for developing a therapeutic formulation to neutralize new emerging VOCs, prevent viral escape, and possibly treat COVID-19 encephalitis.

## 5. Patents

Nbs sequences are published in the pending patent WO 2022/140422.

## Figures and Tables

**Figure 1 viruses-16-00185-f001:**
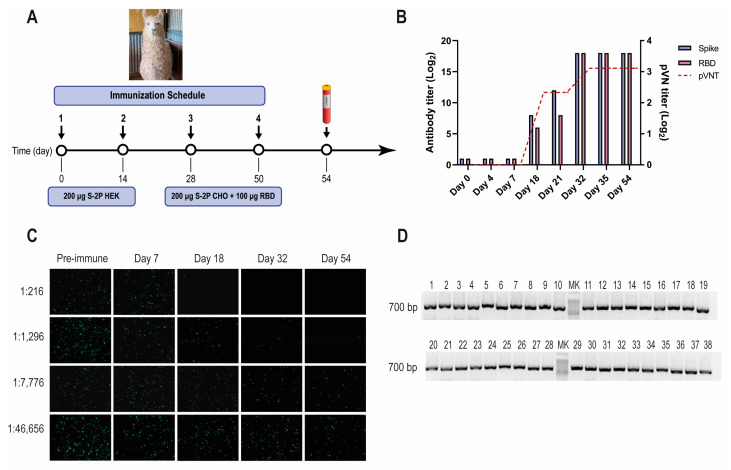
SARS-CoV-2 llama immunization, immune response, and Nb-library construction. (**A**) Immunization schedule: a llama was injected intramuscularly on days 0 and 14 with 200 μg of SARS-CoV-2 S-2P protein produced in HEK-293T, and on days 28 and 56 with 200 μg of SARS-CoV-2 S-2P produced in CHO cells and 100 μg of RBD protein emulsified in Freund’s adjuvant. Four days after the last boost, 200 mL of blood was collected, and peripheral lymphocytes were isolated to produce an immune library; (**B**) Total IgG titer determined by ELISA and neutralizing Ab titer determined by pVNT induced 4 and 7 days after each immunization. Four days after the third immunization (PID 32) a maximal antibody response was reached; (**C**) Picture illustrating neutralizing activity in llama serum determined by pVNT. The neutralization capacity increased after each immunization and correlated with a decrease in the number of fluorescent cells. A higher neutralizing titer was detected for a dilution of 1:1296 at PID 54; (**D**) Analysis of PCR products by agarose gel electrophoresis to confirm the number of transformants that had an insert of the proper size: each of the 48 clones that were randomly selected contained a genuine Nb fragment (~700 bp).

**Figure 2 viruses-16-00185-f002:**
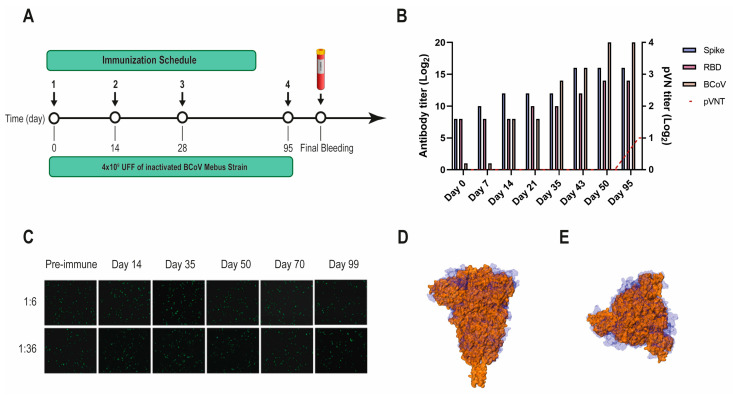
BCoV Mebus llama immunization and immune response. (**A**) Immunization schedule followed to produce the Nb immune library: a llama was injected intramuscularly on days 0, 14, 28, and 95 with 4.00 × 10^9^ UFF of the inactivated BCoV Mebus strain in Freund’s adjuvant. Peripheral lymphocytes were isolated from 200 mL of blood collected four days after the final boost to generate the immune library; (**B**) Total IgG titer determined by ELISA for BCoV and SARS-CoV-2 RBD and S-2P proteins; (**C**) Picture showing non-neutralizing activity of sera from a llama immunized with BCoV Mebus against SARS-CoV-2 determined by pVNT. Superimposition of the S protein structures from SARS-CoV-2 (orange) and BCoV Mebus (blue) using VMD software. Frontal (**D**) and upper (**E**) view.

**Figure 3 viruses-16-00185-f003:**
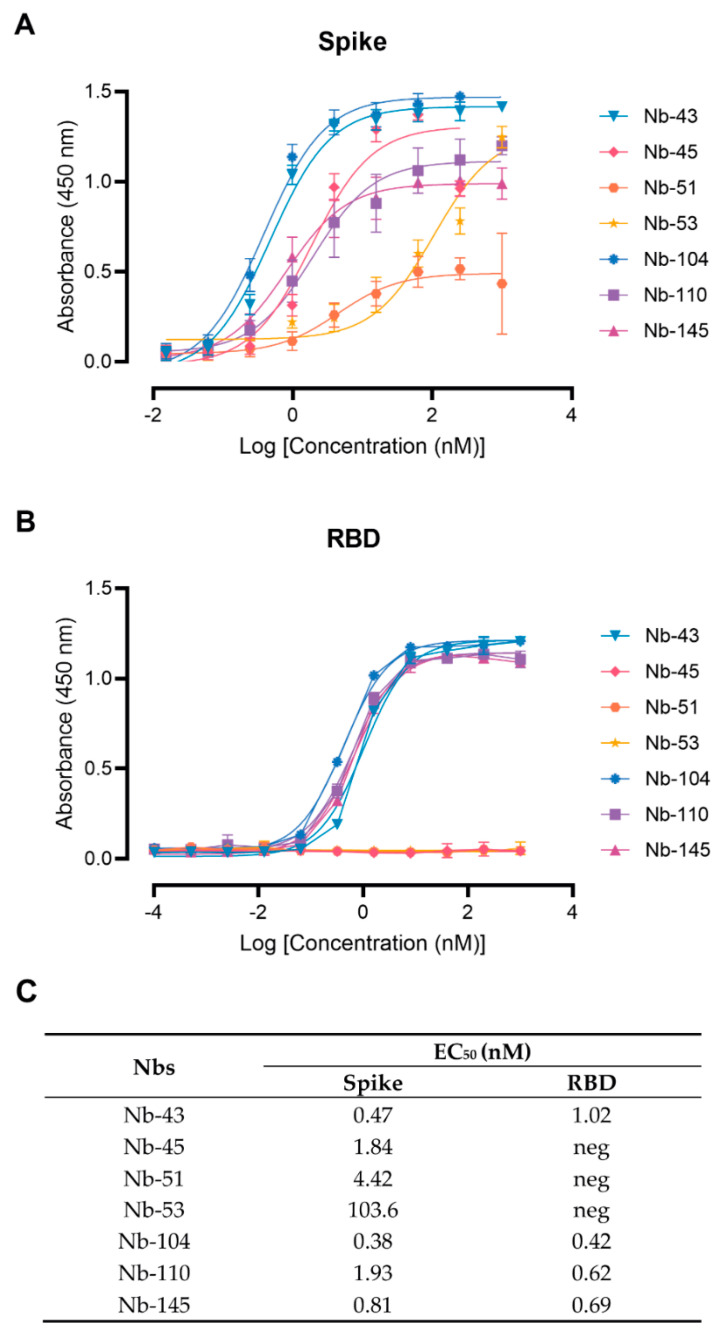
Nanobody affinity to RBD and S-2P from SARS-CoV-2 by ELISA. The binding of the selected Nbs was analyzed in plates coated with S-2P (**A**) and RBD (**B**) proteins from the Wuhan-Hu-1 SARS-CoV-2. Different colors were assigned to each curve according to the Nbs used. Error bars represent the standard deviation (SD) of triplicates; (**C**) Summary table of EC_50_ values against RBD and S domains.

**Figure 4 viruses-16-00185-f004:**
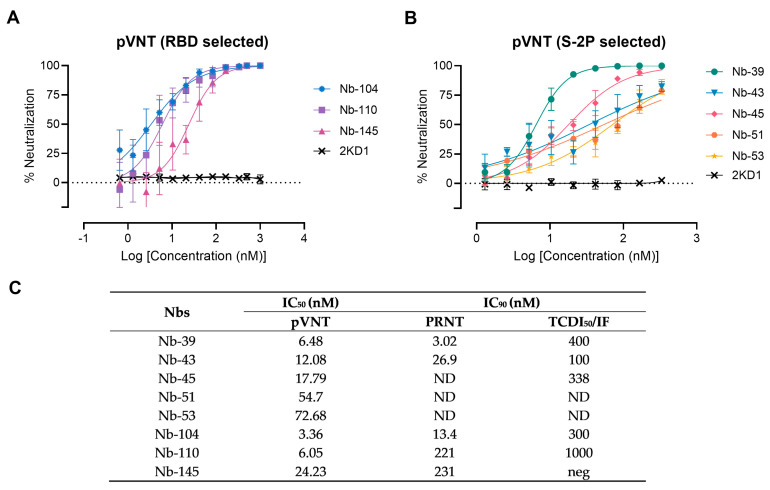
Neutralizing activity of Nbs against SARS-CoV-2 WT strain. The neutralization potency of eight Nbs was calculated based on the pVNT results. (**A**) Blue, violet, and magenta symbols and lines denote Nb-104, Nb-110, and Nb-145 selected with the SARS-CoV-2 RBD protein, respectively; (**B**) Green, light-blue, pink, orange, and yellow symbols and lines represent Nbs selected against the S-2P protein. Black symbols and lines denote an anti-rotavirus control Nb (2KD1). Inhibition curves were performed with the selected Nbs at two-fold serial dilutions. After 48 h, the GFP signal from two images per well was quantified using ImageJ/Fiji and normalized to the number of GFP-positive cells of wells containing only pseudovirus. Inhibition curves are presented in log-transformed dilution with IC_50_ values for each Nb. Each experiment was replicated three times. The IC_50_ was calculated by fitting the inhibition from serially diluted Nbs to a sigmoidal dose–response curve; (**C**) Virus neutralization potency of each Nb was assessed by three different methodologies: pseudovirus neutralization test (pVNT), plaque reduction neutralization test (PRNT), and immunofluorescence assay (IF). Half-maximal inhibitory concentrations IC_90_ were calculated for each Nb in each assay as described in the methods section. ND: not determined.

**Figure 5 viruses-16-00185-f005:**
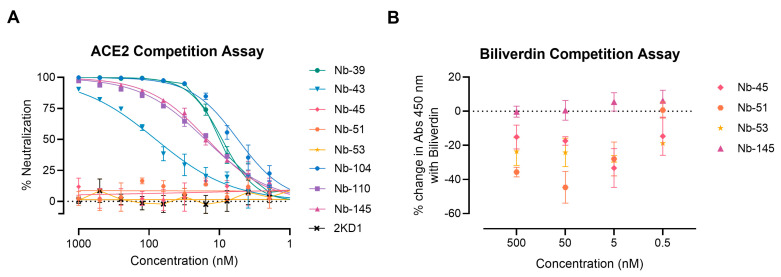
ACE2-RBD and biliverdin competition assays. (**A**) Competitive ELISA of ACE2 binding to RBD immobilized on plates by increasing concentrations of Nbs. The specific binding of ACE2-HRP to RBD was detected with a chromogenic reagent. The IC_50_ was calculated by fitting the inhibition from serially diluted Nbs to a sigmoidal dose–response curve. The experiment was performed in triplicate; (**B**) Competitive ELISA of Nbs at different concentrations binding to S-2P protein in the presence of 5 μM of biliverdin.

**Figure 6 viruses-16-00185-f006:**
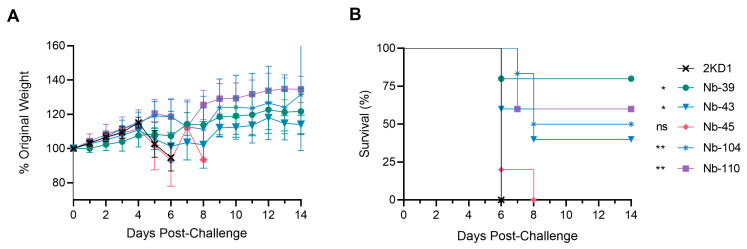
Protection against SARS-CoV-2 challenge in k18-hACE2 mouse model. k18-hACE2 mice were challenged with 1 × 10^5^ PFU of WA-1 SARS-CoV-2 after the intranasal administration of 10 or 20 μg of Nbs. (**A**) The body weight changes in the animals in the control group and treatment groups were recorded for two weeks and compared; (**B**) Survival curves of the different treatment groups. Statistical significance was determined using One-way ANOVA with Dunnett’s post-hoc analysis. * *p* < 0.05, ** *p* < 0.01, ns: not significant.

**Figure 7 viruses-16-00185-f007:**
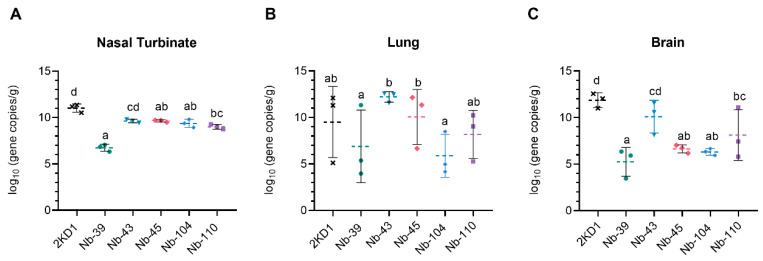
Viral load in different tissues after Nb treatment and virus challenge. Virus load measured by RT qPCR in nasal turbinates (**A**), in the lung (**B**), and the brain (**C**). Tissues were collected at four days after challenge (*n* = 3 mice per tissue, 9 mice per group). Mice were treated with each Nb by the intranasal route and challenged with WA-1 SARS-CoV-2, 4 h later. Samples were prepared from infected mice for RNA isolation and RT-qPCR. Statistical significance was determined using the general mixed 2-way ANOVA model, LSD Fisher’s exact test, and Bonferroni correction. Different letters on top of bars indicate significant differences in viral load among groups for the same tissue (*p* < 0.001), while shared letters indicate no significant difference.

**Figure 8 viruses-16-00185-f008:**
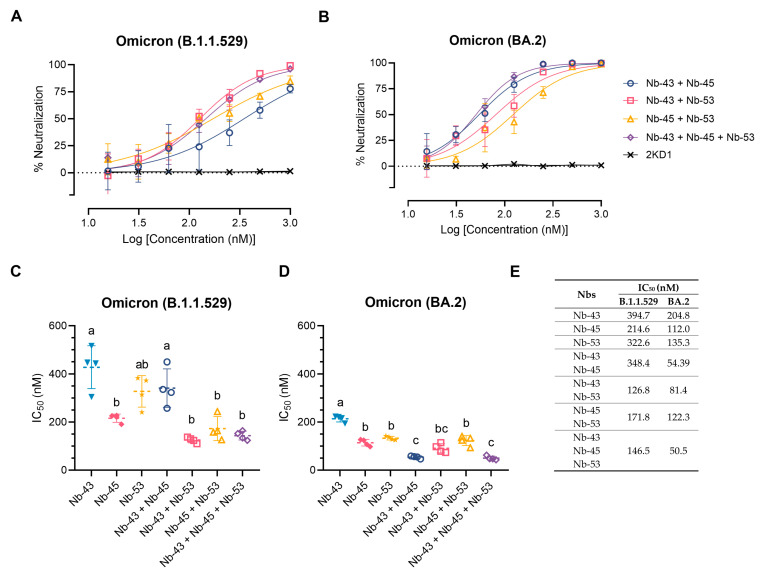
Enhancement of neutralization potencies against Omicron variants using Nbs cocktails. (**A**) Percentage of neutralization exerted by a mixture of two or three Nbs against the B.1.1.529 Omicron variant; (**B**) Similar experiments to A performed with the BA.2 Omicron variant. Comparison of IC_50_ values of Nbs combinations vs. individual Nbs for the B.1.1.529 Omicron variant (**C**) and the BA.2 Omicron variant (**D**). Individual Nbs or Nb-mixtures with different letters differ significantly in IC_50_ (One-way ANOVA, Tukey multiple comparison, *p* < 0.001), (**E**) while shared letters indicate no significant difference.

**Figure 9 viruses-16-00185-f009:**
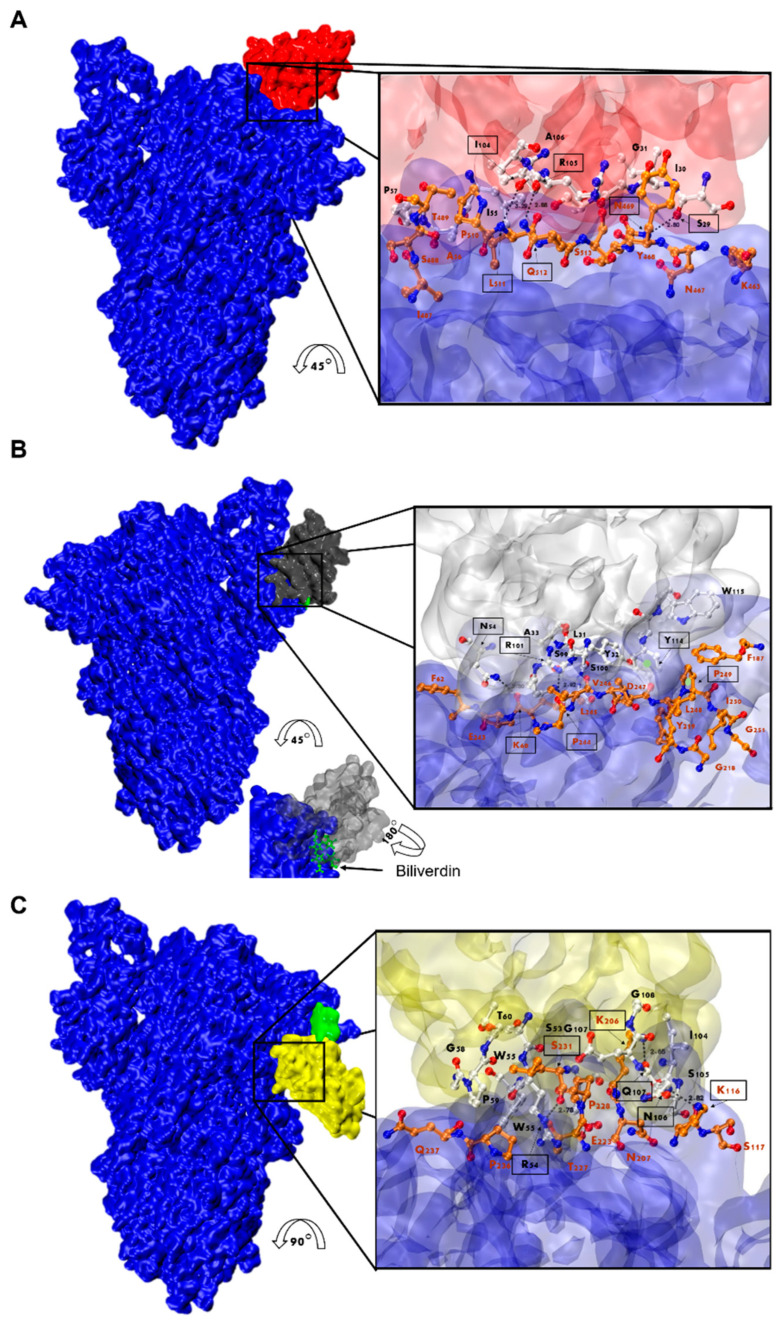
Analysis of Nanobody-S protein interactions. Prediction of the interaction of Nb-43 ((**A**), red), Nb-45 ((**B**), gray), and Nb-53 ((**C**), yellow) with the S protein (blue). Biliverdin is highlighted in green. The three panels display the S-Nb overall view and a close-up view of the interaction zone. The close-up view displays the predicted binding site, with AA in red representing those of the S protein and in black those belonging to the Nb. Amino acids enclosed in a black box and indicated with an arrow show residues that could form hydrogen bonds (dotted lines). Section B shows green circles representing possible Pi-stacking interactions.

**Figure 10 viruses-16-00185-f010:**
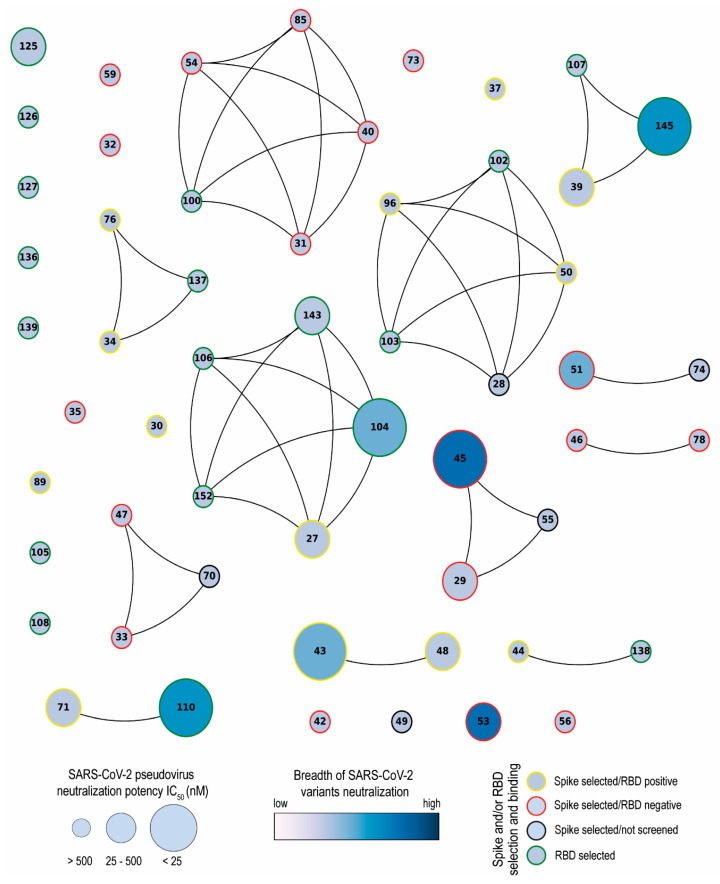
Network analysis of unique nanobodies. A visual representation summarizing the RBD and S-2P binding and neutralization for the 43 isolated Nbs. Nanobodies are depicted as dots in different sizes and colors, and those with a CDR3 sequence identity greater than 70% are connected. Their neutralization potencies against pseudotyped SARS-CoV-2 WT are represented by the size of dots and the filled gradient color represents the breadth of SARS-CoV-2 variant neutralization. Dots are colored on the outer circle based on the antigen used for biopanning and whether they bind to RBD or not.

**Table 1 viruses-16-00185-t001:** Neutralization titers of SARS-CoV-2 strains in vitro by RBD and non-RBD binders.

Nbs		pVNT IC_50_ (nM)						
		Alpha	Beta	Delta	Omicron B.1.1.529	Omicron BA.2	Omicron XBB	Omicron XBB.1.5
RBD binders	Nb-43	-	-	-	394.7	204.8	-	-
	Nb-104	18.29	-	121.8	-	-	-	-
	Nb-110	133.9	324.8	121.8	-	-	-	-
	Nb-145	86.65	451.6	384.4	-	-	-	-
Non-RBD binders	Nb-45	83.66	237.5	248.0	214.6	112.0	30.73	42.17
	Nb-51	215.3	616.3	-	-	-	285.6	316.9
	Nb-53	91.2	112.4	701.8	322.6	135.3	145.2	136.7

## Data Availability

All data are available in this manuscript, its Appendix A, and the patent form. Correspondence and requests for materials should be addressed to L.I.I. (loreitati@gmail.com).

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
