# Peer review of "SARS-CoV-2 Specific Nanobodies Neutralize Different Variants of Concern and Reduce Virus Load in the Brain of h-ACE2 Transgenic Mice"

_viruses, 2024, doi:10.3390/v16020185_

Round 1

Reviewer 1 Report

Comments and Suggestions for Authors

I have read with great interest the article titled “”. This research is profusely developed and well explained.
The objective of the research is well defined. The methods used are well described and explained. Despite the large number of results, they are well developed and the figures are sufficient to fully understand the development of said project.
Finally, the discussion and final conclusions are well presented. The bibliography is correct.

Author Response

We thank the reviewer for his/her time and effort in reviewing our manuscript.

Reviewer 2 Report

Comments and Suggestions for Authors

The manuscript by Pavan and co-workers is very well written; the experiments are clearly explained, the results are adequately explained in general, and the conclusions are consistent with the framework and data. I thank the authors for the care they took to describe the experimental section.

The article contains most of the needed information to appreciate the potential of this treatment fully. However, it would be very informative if they could specifically address is the mice generate Ab against the llamas' Ab. This is a standard point, but it would be very informative for scientists who are not familiar with this system.

I have some minor comments:

1.- There is a mixed use of the comma. Sometimes, it is used to separate decimals, and some other times to separate thousands.

2.- The genus and order should be in italics, not only the viral family.

3.- I strongly suggest to move S1 and S2 to the main text.

4.- In section 2.11, the assays are described as binding kinetics. This term is incorrect as the x-axis is not time but concentration. This is a binding isotherm (equilibrium measurement). These are called binding isotherms.

5.- Please specify the binding isotherm equation used for all non-linear regressions. Where the regressions a 1:1 binding isotherms (Langmuir binding equation) or cooperative binding isotherm?

Major points to be discussed.

1. Figure 5 is barely explained in the section 3.4. Please explain these results better, particularly the fact that there seems to be no inhibition of viral replication in the lower respiratory tract. Also, please address this aspect in the discussion; this result is kind of explained with respect to the ability of the Ab to cross the BBB, but there is no explanation (to the best of my knowledge) for the fact that there was no protection in the lower respiratory tract. Furthermore, figure 4b shows that no treatment was able to achieve a 100% survival rate by day 14. Is this result related to the fact that there is no suppression of viral replication in the lower respiratory tract?

2.- Are there nAb that inhibit SARS-CoV-2 replication in the lower respiratory tract? If so, why do you think your system could not achieve this result?

3.- Given the result from figures 4b and 5b would this treatment be a good option for human patients?

4.- Figure 6 is missing the negative control group.

I have selected the option "Reconsider after a major revision (control missing in some experiments)" because figures 2, 3a 6 are missing the negative controls. Otherwise, the article is phenomenal, but I will kindly ask the authors to address all my points.

Reviewer 3 Report

Comments and Suggestions for Authors

In this manuscript, the authors generated a large immune library by immunizing the llama and screened several nanobodies with strong or broad neutralizing activities against the different VOCs. They provide a clear and complete report on the development and characterization of nanoantibodies. The promising candidates with excellent antiviral capacity enriched the therapeutic reservoirs to combat SARS-COV-2. The manuscript is well structured and well written. I have some minor points to improve the quality of this manuscript.

1.    Line 407-408: Figure 2A-B and their descriptions are inconsistent.

2.    Line 455: Figure S5 doesn't show results on Nb-145.

3.    Typo:

â‘  Line 167/171: I think it should be "2xYT medium", not "2xTY medium".

②Figure S2 legend: 4.00✕109 UFF.

Round 2

Reviewer 2 Report

Comments and Suggestions for Authors

Dear Drs. Ibañez and Parreño,

I greatly appreciate the time and effort you took to address all of my concerns. I think you have written a very good manuscript and I am satisfied with this second version.

all the best

Author Response

Thanks to the reviewer for his/her time